# Habitat Quality Assessment under the Change of Vegetation Coverage in the Tumen River Cross-Border Basin

**Yue Wang** [1], **Donghe Quan** [1], **Weihong Zhu** [1,2], **Zhehao Lin** [1,3] and **Ri Jin** [1,2,3,*]

1   College of Geography and Ocean Sciences, Yanbian University, Yanji 133002, China; 2204271162@ybu.edu.cn (Y.W.); 2021010755@ybu.edu.cn (D.Q.); whzhu@ybu.edu.cn (W.Z.); zhlin@ybu.edu.cn (Z.L.)
2   Jilin Provincial Joint Key Laboratory of Changbai Mountain Wetland & Ecology, Changchun 130102, China
3   Northeast Asian Research Center of Transboundary Disaster Risk and Ecological Security, Yanbian University, Yanji 133002, China
*   Correspondence: jinri0322@ybu.edu.cn; Tel.: +86-130-0908-3971

**Abstract:** The continuous deterioration of terrestrial ecosystems has led to the destruction of many biological habitats in recent years. The Tumen River cross-border basin, an important biological habitat, is also affected by this changing situation. Assessing habitat quality (HQ) is crucial for restoring and protecting habitats, and vegetation plays a significant role in this process. In this study, we used geographical detector (GD) to extract fraction vegetation coverage (FVC) features and quantify the contribution of driving factors. By coupling vegetation cover and land use data, we assessed HQ. Our findings reveal a declining trend in FVC from 2000 to 2020, which mainly assumed a spatial pattern inclined from northeast and southwest to southeast. Human activities and natural factors interacted to cause these changes in FVC, with human activities having a more significant impact. Vegetation and land use changes led to a decline in the basin's HQ index. This study highlights the crucial role of FVC in HQ and provides a relevant scientific reference for optimizing the evaluation of HQ in the Tumen River cross-border basin and promoting the sustainable development of regional ecology.

**Keywords:** fraction vegetation coverage; habitat quality; geographical detector; Tumen River cross-border basin

## 1. Introduction

Habitat quality (HQ) is a crucial indicator of an ecosystem's suitability for the survival and persistence of individuals and populations [1], embodying fundamental attributes of the ecological environment. Since the industrialization revolution, human activities have increasingly disturbed terrestrial ecosystems [2], leading to a reduction in and the degradation of ecosystem service capacities, intensifying ecological crises, and threatening the ecological security pattern [3,4]. Fraction vegetation coverage (FVC) is the ratio of the vertical projected area of the vegetation canopy, branches, and leaves on the ground of a growth area to the total area of the statistical area [5,6]. It can be used to accurately monitor the dynamic changes of vegetation in time and space and indicates changes in a regional ecological environment [7,8]. Therefore, analyzing the quality of regional habitats based on changes in FVC is of great significance for scientifically planning regional ecological environments [9].

Scholars have been monitoring FVC for a long time. Initially, surface FVC was mainly monitored through field measurements. Although the data obtained were highly accurate, the measurement process was time-consuming and laborious. The differences in the selection of different points increased the uncertainty of the FVC measurements and made it difficult to better reflect the dynamics of large-scale vegetation coverage. However, with the rapid development of earth observation technology and the onset of the big

data era in earth science, scholars began to apply remote sensing data to FVC monitoring, leading to the development of many remote sensing estimation methods for determining FVC [10–13]. The remote sensing estimation method has the advantages of wide coverage, strong temporal and spatial continuity, and the ability to easily realize regional-scale conversion [14], thereby compensating for the limitations of field measurements. In recent years, some scholars have extensively studied FVC changes in different regions using the dimidiate pixel model [15–17]. It is evident that the use of remote sensing technology can accurately capture the status of surface vegetation in real time, which is essential for the protection of vegetation communities [18].

HQ serves as an indicator of regional biodiversity levels [19]. This quality, along with its fluctuations, is influenced by a multitude of factors, encompassing both natural processes and human activities. Human-induced changes in land use patterns greatly impact regional species and their habitats [20,21]. The integrated valuation of ecosystem services and tradeoffs (InVEST) model offers a comprehensive assessment of HQ via considering the influence of human activities, utilizing land use data, and incorporating habitats' sensitivity to threats [22,23]. Beyond land use changes, a holistic evaluation of habitat quality necessitates the integration of various ecosystem attributes, such as biological species, vegetation, and other elements [24]. Researchers have endeavored to quantify these data through field surveys. However, large-scale studies are hindered by high costs and limited time spans. Vegetation plays a significant role in the material and energy cycles between habitat patches [24] and is significantly related to HQ [25]. Vegetation type, structure, and growth status affect species habitat selection [26,27] and can quantify the suitability of the habitat in question [28]. To address these challenges, this study leverages the advantages of long-term satellite observations, such as their macroscopic perspectives, rapid data acquisition capacity, and real-time capabilities, and applied FVC, as a representative indicator of vegetation, to the HQ index, thus enabling a more comprehensive and scientific evaluation of HQ in the Tumen River cross-border basin.

Spatial analysis of data is a pivotal aspect of research, particularly with respect to understanding the intricacies of spatial effects [29]. In practical surveys, HQ data are often influenced by various environmental factors, exhibiting noticeable spatial heterogeneity based on geographical location [30]. Traditional statistical approaches tend to overlook the spatial variation inherent in natural phenomena, masking local discrepancies among variables and yielding potentially biased estimations of spatial distribution. Geostatistics, which is rooted in the theory of regionalized variables, provides a robust framework for analyzing spatial heterogeneity and correlations in natural phenomena [31]. Its application in ecological research has proven effective in describing spatial data and quantifying uncertainties [32–35]. To investigate the spatial characteristics of the analyzed data [36], this study employs a geographically weighted regression (GWR) tool to elucidate the spatial patterns of habitat quality and its associated variables.

The Tumen River Basin, located in the core region of the Northeast Asian ecological network, possesses rich animal and plant resources and plays a crucial ecological role. The basin's development has accelerated with the increasing degree of international cooperation between China, North Korea, and Russia, leading to the degradation of key regional ecosystems and intensifying ecological crises. Scholars have been attracted to the eco-environmental issues concerning the Tumen River cross-border basin, resulting in many cooperative and research efforts [37–39]. However, the Tumen River Basin is located in the cross-border area between China and North Korea. Due to regional constraints, the research conditions are poor, and the corresponding research is difficult. Therefore, there are few scientific studies on the overall FVC and HQ in the Tumen River cross-border basin. Therefore, using these considerations as a starting point, this study discusses the entire Tumen River cross-border basin, both the Chinese side and the Korean side, at three scales so as to provide important data support for the study of the HQ of the Tumen River cross-border basin.

Based on MOD13Q1 NDVI obtained data from 2000 to 2020, this study employs the dimidiate pixel model method to evaluate FVC in the Tumen River cross-border basin. The temporal and spatial variation characteristics of FVC are analyzed using the one-variable linear trend method. The geographical detector (GD) model is used to quantify the contribution rate of each influencing factor to FVC and determine its suitable range. The InVEST model is used to calculate the HQ index of the Tumen River Cross-border Basin in the period from 2000 to 2020. Using the geographically weighted regression (GWR) tool, the change in the HQ index is used as the dependent variable, and the change in FVC is used as the independent variable to quantify the impact of FVC on HQ. Moreover, the paper couples vegetation coverage and land use data based on the InVEST model to more accurately and scientifically evaluate the HQ of the Tumen River cross-border basin. The study aims to provide a scientific theoretical basis and data support for the ecological environmental planning of the Tumen River Basin and the coordination of nature, economy, and society towards realizing green and sustainable development.

## 2. Materials and Methods

### 2.1. Study Area

The Tumen River cross-border basin, located at the junction of China, North Korea, and Russia, covers approximately 33,170.41 km$^2$, with the Chinese side accounting for about 68%, including most of the eastern part of Yanbian Korean Autonomous Prefecture (Figure 1). The North Korean side accounts for 31% and covers Ryanggang Province and North Hamgyong Province, while a small portion of the estuary belongs to the Hassan district of Russia's Primorsky Krai. The basin lies in the middle temperate zone of the northern hemisphere and is significantly influenced by vapor from the Sea of Japan and the southeast monsoon, leading to a humid monsoon climate. The annual average temperature ranges from 2–6 °C, while the annual precipitation level is around 400–800 mm, which is mainly concentrated in the summer, resulting in good water and heat conditions and a healthy forest coverage rate. The vegetation coverage in this area is mostly coniferous forest and mixed coniferous–broadleaved forest. The main tree species include larch (*Larix gmelinii* Rupr. Kuzen.), fish scale spruce (*Picea jezoensis* var. microsperma), and birch (*Betula platyphylla* Suk.) [40]. The basin is home to a diverse range of natural ecosystems harboring numerous rare animals and plants, such as the *Red-crowned crane* (*Grus japonensis*), *Amur tiger* (*Panthera tigris* ssp. *altaica*), *Korean pine* (*Pinus koraiensis* Sieb. et Zucc.), *Korean fir* (*Abies koreana* E.H.Wilson), and many others. Various landforms are present in the area, with terrain higher in the south and lower in the north and sloping from southwest to northeast. Owing to the interaction between nature and human activities, FVC patterns in the basin exhibit significant temporal and spatial variations, with changes in FVC impacting the integrity of the habitat.

### 2.2. Data Source and Processing

The data and sources used in this study are listed in Table 1.

The MOD13Q1-MODIS/Terra Vegetation Indices 16-Day L3 Global 250 m SIN Grid data were downloaded from the National Aeronautics and Space Administration (NASA) website (Table 1) [41], and the Maximum Value Composites (MVC) method [42] was used to synthesize annual vegetation index data. This approach effectively mitigated noise interference such as cloud shadows and sun height [43]. The meteorological data used for this study were obtained from Version 4 of the CRU TS monthly high-resolution gridded multivariate climate dataset provided by the Climatic Research Unit (CRU) (Table 1) [44], and the surface temperature data were provided by the National Oceanic Atmospheric Administration Physical Sciences Laboratory (Table 1) [45]. Annual data synthesis was carried out to derive the annual average temperature, annual average surface temperature, and annual precipitation data of the Tumen River Basin from 2000 to 2020, which were downscaled to 250 m using the inverse distance weighting method. The DEM data used were ASTER GDEM data, which were obtained from the geospatial data cloud platform

(Table 1). Slope and aspect data were calculated using ArcGIS Pro. Population density data were obtained from the LandScan Global dataset with a spatial resolution of 30″ (Table 1), while the land use data for the three periods of 2000, 2010, and 2020 were provided by GLOBELAND30 with a spatial resolution of 30 m (Table 1). Night light satellite data, covering the period from 2000 to 2020, were obtained from the extended time series of global NPP-VIIRS-like nighttime light data acquired through across-sensor calibration (Table 1) [46]. Other auxiliary data, such as those concerning railways, highways, and settlements, were obtained from China's 1:1 million public version of the basic geographic information database (2021) (Table 1). For ease of analysis, the data were uniformly reprojected to WGS_1984_UTM_Zone_52N and resampled to 250 m × 250 m.

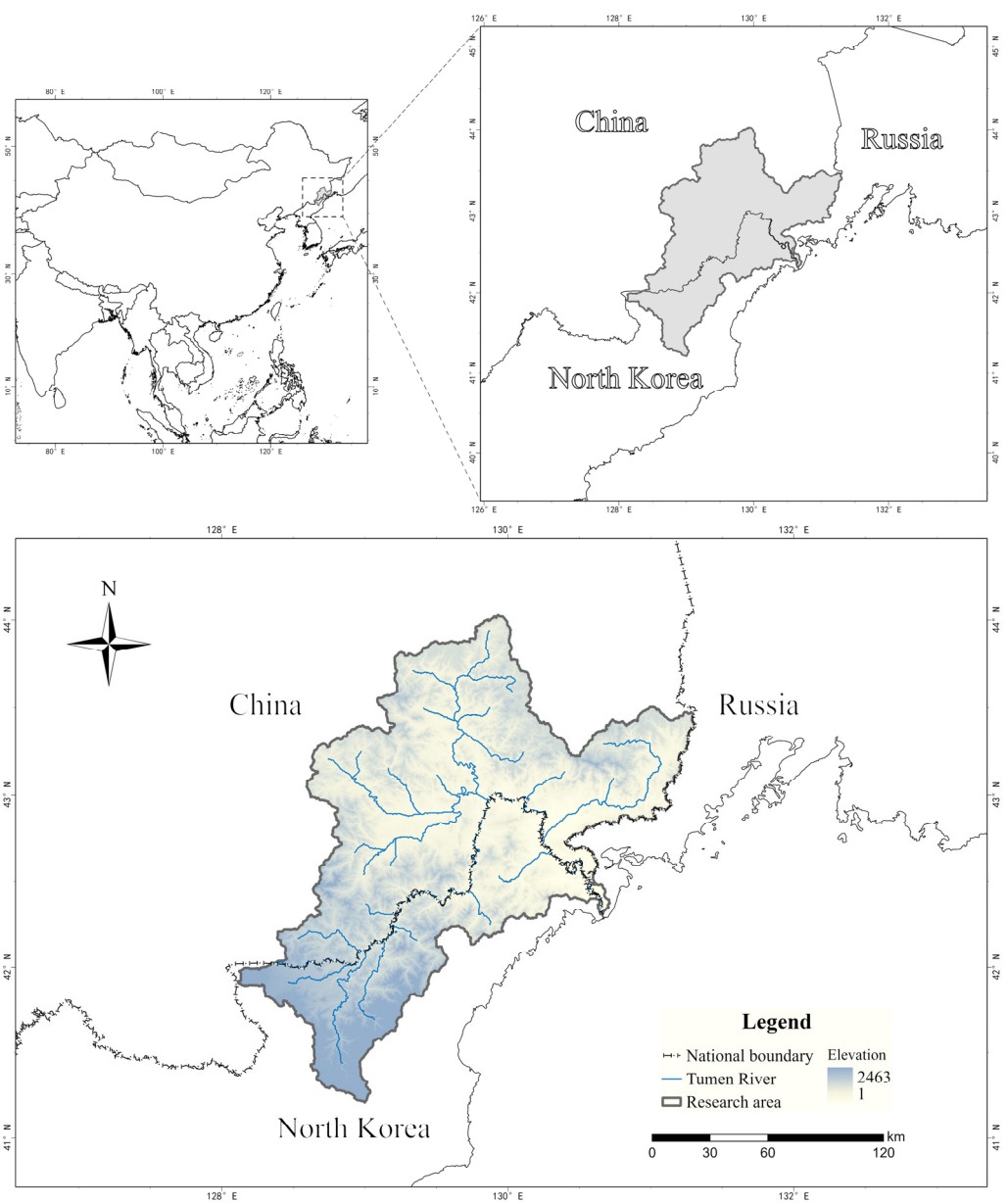

**Figure 1.** Location and digital elevation model of the study area.

### 2.3. Methodology

Initially, we collected data on vegetation coverage of the Tumen River cross-border basin, such as remote sensing image, statistical, and survey data. We then built a regional data analysis and processing platform to determine the FVC in the study area alongside other natural and human-activity-related data. By referencing the literature and using the univariate linear trend

method and other technical means, we analyzed and studied the spatio-temporal variation characteristics of FVC in the Tumen River cross-border basin from 2000 to 2020 at the global, Chinese, and North Korean scales. Additionally, we used GD to explore and measure the driving factors that caused this dynamic change process. Subsequently, we employed the InVEST habitat quality model to obtain the habitat quality index of the study area based on the land cover data. By combining the analytical results of dynamic changes in FVC, we quantitatively assessed the impact of these changes on habitat quality in the Tumen River cross-border basin from 2000 to 2020 based on the GWR model system. Finally, we systematically and quantitatively evaluated the joint effect of vegetation status and human activities on the habitat quality change in the Tumen River cross-border basin and used the Moran's I method to analyze and discuss its agglomeration characteristics and spatial pattern.

**Table 1.** Basic data used in the study.

| Type | Name | Code | Resolution | Source |
|---|---|---|---|---|
| Vegetation | Normalized Digital Vegetation Index | NDVI | 250 m | MOD13Q1-MODIS/Terra Vegetation Indices 16-Day L3 Global 250 m SIN Grid provided by NASA (https://ladsweb.modaps.eosdis.nasa.gov/, accessed on 11 February 2023) |
| Climatic | Monthly mean air temperature | Tem | 0.5° | CRU TS v. 4.06 provided by the CRU (https://crudata.uea.ac.uk/, accessed on 11 February 2023) |
| | Monthly total of precipitation | Pre | | |
| | Monthly mean land surface air temperature | Lst | | GHCN_CAMS Gridded V2 by the NOAA Physical Sciences Laboratory (https://psl.noaa.gov/, accessed on 11 February 2023) |
| Topographic | Elevation | Ele | 30 m | ASTER GDEM 30 m (https://www.gscloud.cn/, accessed on 11 February 2023) |
| | Slope | Slo | | |
| | Aspect | Asp | | |
| Human activity | Density of population | Pop | 30″ | LandScan Global population dataset (https://landscan.ornl.gov/, accessed on 11 February 2023) |
| | Land use and land cover | LULC | 30 m | GLOBELAND30 (http://www.globallandcover.com/, accessed on 11 February 2023) |
| | Nighttime light satellite data | Light | 15″ | An extended time series (2000–2020) of global NPP-VIIRS-like nighttime light data from across-sensor calibration (https://doi.org/10.7910/DVN/YGIVCD, accessed on 11 February 2023) |
| | Railway | Railway | / | China 1:1 million public version basic geographic information database (2021) (https://www.webmap.cn/, accessed on 11 February 2023) |
| | Highway | Highway | | |
| | Settlement | Settlement | | |

Note: The railway, highway, and settlement data are all vector data that have been converted into 250 m × 250 m grids and used as stress factors in the assessment of HQ.

### 2.3.1. Dimidiate Pixel Model

The normalized difference vegetation index (NDVI) can provide an objective measurement of vegetation growth conditions [47]. FVC is closely related to NDVI and can accurately quantify vegetation dynamics [48]. The dimidiate pixel model is based on NDVI data and hybrid pixel

decomposition theory. Using the conversion relationship between NDVI and FVC, this model can accurately calculate FVC. The calculation formula is as follows [49]:

$$FVC = \frac{NDVI - NDVI_{soil}}{NDVI_{veg} - NDVI_{soil}} \quad (1)$$

where $NDVI_{soil}$ represents the NDVI value of pure bare-land pixels and $NDVI_{veg}$ represents the NDVI value of pure vegetation pixels. The values selected for $NDVI_{soil}$ and $NDVI_{veg}$ are influenced by atmospheric conditions, vegetation types, and other factors. To minimize noise interference from factors such as cloud shadow and sun height, this study utilized the MVC method to synthesize the annual maximum NDVI. The values corresponding to the 5th and 95th percentiles of the annual maximum NDVI cumulative frequency were used as $NDVI_{soil}$ and $NDVI_{veg}$, respectively [12]. Based on the actual conditions of the Tumen River Basin, FVC can be classified into five categories: very low vegetation coverage (<0.15), low vegetation coverage (0.15–0.35), medium vegetation coverage (0.35–0.55), high vegetation coverage (0.55–0.75), and very high vegetation coverage (>0.75) [8].

### 2.3.2. Univariate Linear Trend

Determining the univariate linear trend is a useful method for estimating the change trend of each pixel over time, allowing for the detection of changes in FVC over a given time frame and providing information on the spatial evolution of FVC [50,51]. The slope of a trend line can be calculated using the following formula:

$$\theta_{slope} = \frac{n\sum_{i=0}^{n}\left(i \times FVC_i\right) - \sum_{i=0}^{n}i \times \sum_{i=0}^{n}FVC_i}{n\sum_{i=0}^{n}i^2 - \left(\sum_{i=0}^{n}i\right)^2} \quad (2)$$

where $\theta_{slope}$ is the slope value of the trend line, n is the number of years for long-term series analysis, and $FVC_i$ is the FVC in the i-th year. $\theta_{slope}$ reflects the slope of the FVC, where a positive value indicates an increasing FVC over time, and a negative value indicates a decreasing trend. The significance of the change trend was verified using the F-test, considering both the slope of the trend line and the significance level. The FVC change trend in the Tumen River basin was classified into five categories: extremely significant degradation ($\theta_{slope} < 0$, $p \leq 0.01$), significant degradation ($\theta_{slope} < 0$, $0.01 < p \leq 0.05$), no significant change ($\theta_{slope} < 0, p > 0.05$ or $\theta_{slope} > 0$, $p > 0.05$), extremely significant improvement ($\theta_{slope} > 0$, $p \leq 0.01$), and significant improvement ($\theta_{slope} > 0$, $0.01 < p \leq 0.05$).

### 2.3.3. Geographic Detector Model

The geographic detector model enables the detection of spatial heterogeneity of geographical phenomena [52]. It comprises four detectors, namely, a factor detector, interactive detector, risk detector, and ecological detector [53]. This tool can expose the spatial associations of geographic elements and explore the contribution rate of evaluation indicators to geographic elements [52]. Surface vegetation is affected by various natural and human factors. Thus, this study has selected nine evaluation indicators, including Tem, Pre, Lst, Ele, Slo, Asp, LULC, Pop, and Light, to ascertain their contributions to vegetation coverage and the interactions among these indicators.

### 2.3.4. Habitat Quality Assessment

This study uses the HQ model of InVEST to assess the HQ of the Tumen River cross-border basin [54]. This model evaluates HQ by establishing a connection between land cover types and stress factors and assessing the degree of impact of these stress factors on habitats [55]. The resulting HQ index is a continuous variable ranging from 0 to 1, for

which higher values indicate greater availability of living resources [56]. The corresponding calculation formula is as follows:

$$Q_i = H_j \left( 1 - \left( \frac{D_{xj}^z}{D_{xj}^z + k^z} \right) \right) \tag{3}$$

where $Q_i$ represents the HQ assessed using the InVEST model, $H_j$ is the habitat suitability of land cover type j, $D_{xj}$ is the threat to land cover type j, Z is a scaling parameter reflecting spatial heterogeneity, and k is the half-saturation parameter (usually assuming a value of 0.05). The values of the maximum impact distance, weight, and distance attenuation function of each stress factor (Table 2) and the habitat suitability of different land cover types and their sensitivity to stress factors (Table 3) were determined based on the relevant literature and the actual conditions of the Tumen River basin [57–60].

**Table 2.** Attribute table of stress factors.

| Threat | Max_Dist | Weight | Decay |
|---|---|---|---|
| Artificial Surfaces | 3 | 1 | exponential |
| Cultivated Land | 1 | 0.7 | linear |
| Railway | 4 | 0.6 | exponential |
| Highway | 3 | 0.5 | exponential |
| Settlement | 2 | 0.4 | linear |

**Table 3.** Habitat suitability of different land cover types and their sensitivity to threat factors.

| Land Cover Type | Habitat Suitability Index | Sensitivity of Habitat Types to Each Threat | | | | |
| | | Artificial Surfaces | Cultivated Land | Railway | Highway | Settlement |
|---|---|---|---|---|---|---|
| Cultivated Land | 0.4 | 0.5 | 0.3 | 0.3 | 0.2 | 0.5 |
| Forest | 0.8 | 0.5 | 0.4 | 0.6 | 0.5 | 0.3 |
| Grassland | 0.6 | 0.6 | 0.5 | 0.7 | 0.6 | 0.4 |
| Wetland | 0.9 | 0.7 | 0.6 | 0.8 | 0.7 | 0.5 |
| Water Bodies | 0.8 | 0.8 | 0.7 | 0.7 | 0.6 | 0.6 |
| Artificial Surfaces | 0 | 0 | 0 | 0 | 0 | 0 |
| Bare land | 0.1 | 0.3 | 0.1 | 0.2 | 0.2 | 0.1 |

### 2.3.5. Geographically Weighted Regression Model

GWR is a spatial analysis technique that considers the influence of observations at various spatial locations on the estimation of regression point parameters [61]. By incorporating spatial location information, this model extends the traditional regression framework, thereby reflecting the non-stationarity of parameters in different spaces, producing results that are more aligned with objective reality [62]. To simplify data and reduce loss of accuracy, the study area was divided into 2 km × 2 km grids [63]. The GWR model utilized the change in HQ index from 2000 to 2020 as the dependent variable and the change in FVC as the independent variable. The GWR tool of ArcGIS Pro was used to model this relationship, and the Gauss function was adopted as the space weight function. The optimal bandwidth was selected based on the principle of minimum Akaike information criterion (AIC) value. The calculation formula is as follows:

$$y_i = \beta_0(u_i, v_i) + \sum_{j=1}^{P} \beta_j(u_i, v_i) x_{ij} + \varepsilon_i \tag{4}$$

where $y_i$ represents the fitted value of the HQ index change for sample i, $x_{ij}$ represents the value of the j-th independent variable for sample i, and the coordinates for the target area i are $(u_i, v_i)$. $\beta_0(u_i, v_i)$ represents the estimated constant value for sample i, while $\beta_j(u_i, v_i)$

represents the local estimated coefficient for independent variable $x_{ij}$. Finally, $\varepsilon_i$ represents the error term, which follows an independent normal distribution.

### 2.3.6. Improved InVEST Habitat Quality Model

In order to accurately assess the status of HQ, it is necessary to integrate multiple attributes of the ecosystem, as solely relying on land use classification data may not suffice. Vegetation plays a great role in habitat selection for many species and can provide insight into the current suitability of the analyzed habitat, which is closely linked to HQ. Thus, this study aimed to improve the InVEST model by incorporating FVC [64], which allowed for a more comprehensive evaluation of HQ. By incorporating FVC into the HQ assessment framework, the impact of FVC on habitat change can be better highlighted. The improved HQ assessment formula is as follows:

$$Q_x = Q_i \times Q_f \tag{5}$$

where $Q_x$ is the comprehensive HQ, $Q_i$ is the HQ assessed by the InVEST model, and $Q_f$ is the FVC.

### 2.3.7. Spatial Autocorrelation Analysis

Spatial autocorrelation analysis is used to describe the aggregation characteristics and spatial pattern of attribute values across entire regions [65]. This type of analysis is divided into two categories: global autocorrelation and local autocorrelation. Global autocorrelation can be used to measure the degree of spatial agglomeration of attribute values across an entire region, while local autocorrelation can further identify the spatial location of the agglomeration center and any abnormal points. Two primary methods are used to conduct these analyses: Moran's I and Getis-Ord G. In this study, we used Moran's I method to quantify the spatial characteristics of the HQ index in the Tumen River cross-border basin [66]. The calculation formula is as follows:

$$\text{Global Moran's I} = \frac{n\sum_{i=1}^{n}\sum_{j=1}^{n}W_{ij}(x_i - \bar{x})(x_j - \bar{x})}{\sum_{i=1}^{n}\sum_{j=1}^{n}\sum_{i=1}^{n}(x_i - \bar{x})^2} \tag{6}$$

$$\text{Local Moran's I}_i = \frac{(x_i - \bar{x})}{\sum_{i=1}^{n}(x_i - \bar{x})^2}\sum_{j}^{n}W_{ij}(x_j - \bar{x}) \tag{7}$$

where n represents the number of spatial positions, $x_i$ and $x_j$ represent the attribute values of spatial positions i and j, and the spatial weight $W_{ij}$ denotes the proximity relationship between spatial positions i and j. Global Moran's I ranges from −1 to 1. A value greater than zero indicates positive spatial autocorrelation in the region as a whole, and the degree of agglomeration increases with a larger value. A value less than zero indicates negative spatial autocorrelation in the entire region, and the degree of dispersion increases with a smaller value. When the value is equal to zero, this indicates high randomness or no spatial autocorrelation. If the value of Local Moran's I is greater than zero, it indicates that the spatial unit has properties similar to its adjacent units, that is, local aggregation. On the other hand, if its value is less than zero, it indicates local dispersion.

## 3. Results

### 3.1. Temporal and Spatial Distribution of Fraction Vegetation Cover Changes in the Tumen River Cross-Border Basin

Based on the temporal trends in FVC in the Tumen River cross-border basin between 2000 and 2020 (Figure 2), it was observed that the Chinese side generally had higher FVC than the North Korean side. The years with low FVC values in the entire basin were 2005 and 2019, whose corresponding values were 0.5997 and 0.6003, respectively. The FVC

in the basin showed a relatively obvious downward trend from 2000 to 2005. From the perspective of the entire basin, the speed of the FVC decline was −0.0082/10a, and the R-squared value was 0.5254. From 2005 to 2010, the FVC of the whole basin and the Chinese side increased, with corresponding speeds of 0.0064/10a and 0.0040/10a, respectively. Although the FVC on the North Korean side has declined, the vegetation in the Tumen River cross-border basin is still gradually recovering. The FVC in 2010 was lower than that in 2000, indicating a decline from 2000 to 2010. From 2010 to 2020, the FVC in each region of the Tumen River cross-border basin fluctuated greatly but showed a slight downward trend overall. In summary, the FVC in the Tumen River cross-border basin displayed a general downward trend from 2000 to 2020.

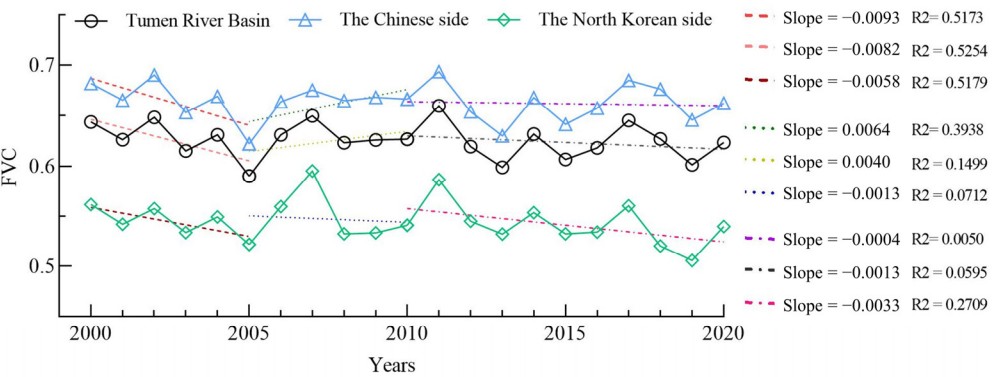

**Figure 2.** Temporal trends of FVC in the Tumen River cross-border basin from 2000 to 2020.

The distribution map of FVC in the Tumen River cross-border basin for the four periods (Figure 3) reveals that the overall FVC in the study area is relatively high, with the main types of FVC being high and very high. The low FVC values are mainly concentrated in the middle of the basin and a small area in the southwest, including the plains, hills, and river valleys with minor terrain fluctuations. The high value agglomeration areas are located in the north and west of the basin. The spatial pattern of FVC in the study area exhibits a trend of inclination from the northwest and northeast to the southeast.

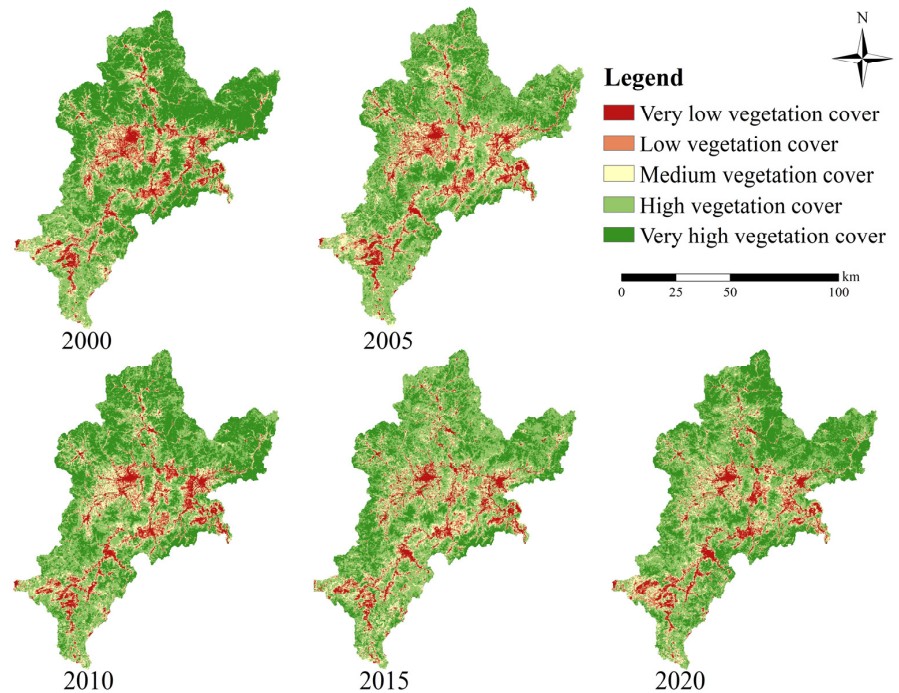

**Figure 3.** Spatial distribution of FVC in the Tumen River cross-border basin from 2000 to 2020.

Based on the vegetation coverage area ratios between 2000 and 2020 (Figure 4), the area of very high vegetation coverage follows a 10-year cycle of decline and rise. However, the overall trend of a very high vegetation coverage area is decreasing due to a decline in FVC, with the proportion dropping from 44.06% to 35.73%. The cycles of change for high and medium vegetation coverage are also 10 years, with both showing a trend of initially rising and then falling, and they are mostly transformed with adjacent vegetation cover types. In addition, the medium- and low-vegetation-coverage types are concentrated around the cultivated land and border areas with forest land, and their areas fluctuate greatly as a result of human activities. The area of very low vegetation coverage, which is primarily distributed around urban residential land, experienced minimal fluctuation from 2000 to 2020. Although urban expansion has occurred, it has not been significant due to the small proportion of this type of land use with respect to the total area.

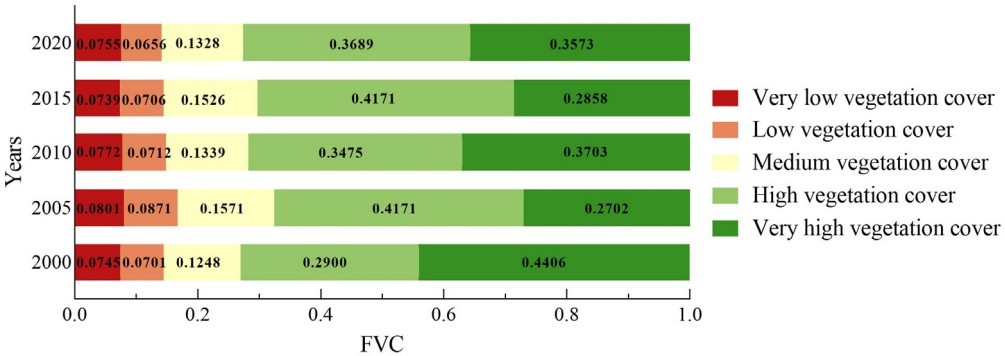

**Figure 4.** Percentage of vegetation coverage types in the Tumen River cross-border basin from 2000 to 2020.

The transfer and change of vegetation cover types in the Tumen River cross-border basin exhibit significant temporal and spatial differences (Figure 5). During the period of 2000 to 2005, the transfer area of the extremely high vegetation cover type on the Chinese side was the largest, covering an area of about 5547 km$^2$. Most of the transferred areas were transformed from very high vegetation coverage to high vegetation coverage, accounting for 94.38% of the transfer area, followed by high vegetation coverage, which was mostly converted to medium vegetation coverage. The North Korean side also has the largest conversion area of extremely high vegetation coverage, followed by high vegetation coverage; both are mainly converted to lower-level vegetation cover types. From 2005 to 2010, both the Chinese side and the North Korean side had the largest transfer area of high vegetation coverage. In addition, both have the largest conversion area to the extremely high vegetation coverage type, namely, 4539 km$^2$ and 1251 km$^2$, respectively, accounting for 48.95% and 30.14% of the high-vegetation-coverage type, and the transfer rate is relatively large.

During the 2010–2015 period, both the Chinese and North Korean sides experienced significant mutual conversion between the very high vegetation coverage type and the high vegetation coverage type, with the largest transfer area corresponding to the former category. Moreover, the transfer areas of both regions with low vegetation coverage accounted for the largest proportions of their respective areas at 47.84% and 46.19%. From 2015 to 2020, the Chinese side saw the largest transfer area of the high-vegetation-coverage type, which mainly corresponded to a transfer to the very-high-vegetation-coverage type. Notably, the area of very high vegetation coverage increased by about 1900 km$^2$ compared to 2015. Meanwhile, the ratio of low vegetation coverage transfer area to its own area was the highest, with a shift to medium vegetation coverage. The transfer-out rate of the very-low-vegetation-coverage type was the smallest, remaining relatively stable. On the North Korean side, there was a significant increase in FVC from medium vegetation coverage to high vegetation coverage, accounting for an area of about 1171 km$^2$. Addi-

tionally, the largest transfer area was observed from high vegetation coverage to very high vegetation coverage.

The Chinese side

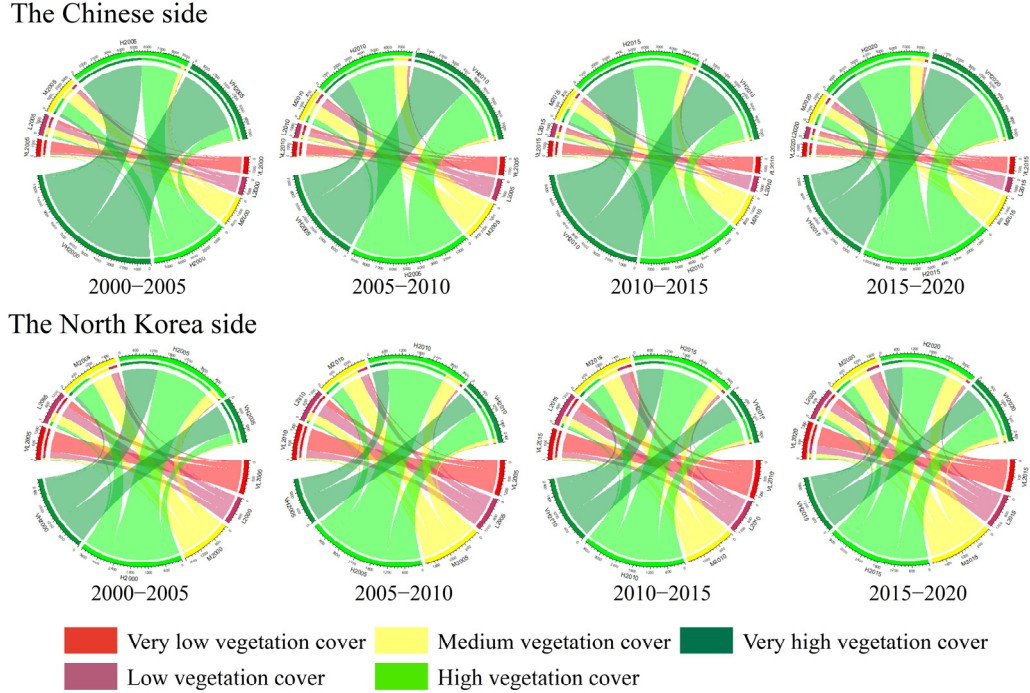

The North Korea side

**Figure 5.** Changes of vegetation coverage types in the Tumen River cross-border basin from 2000 to 2020.

To investigate the spatio-temporal changes in FVC, we estimated the change trend of FVC pixel by pixel from 2000 to 2020 using linear trend analysis. This helped us identify the dynamic changes in FVC in the Tumen River Basin during five different periods (Figure 6). The results reveal that the northern part of China and the mountainous areas of North Korea exhibited relatively active changes in FVC during the four periods from 2000 to 2020. The area wherein FVC remained relatively stable amounted to about 25,886.23 km$^2$, accounting for approximately 79.96% of the Tumen River cross-border basin. Around 11.94% of the areas showed a degraded trend of vegetation, which was slightly higher than the proportion of areas where vegetation improved. The areas with improved FVC were mainly located in the middle of the study area, primarily in the surrounding areas of farmland. However, the mountainous areas in the northwest, southeast, and northeast of the study area, including Yanji City, Helong City, and Hunchun City in Yanbian Korean Autonomous Prefecture on the Chinese side and the northeastern part of North Hamgyong Province and the Hoeryong area of Ryanggang Province on the North Korean side, showed degrading FVC.

*3.2. Analysis of Driving Factors of Vegetation Cover Change in the Tumen River Cross-Border Basin*

The interaction between human activities and natural factors has influenced the pattern and change of land surface vegetation. GD can measure the degree of influence of each factor on FVC. The results from factor detector analysis (Q Value) showed that Pop (0.4779) followed by LULC (0.4269) are the strongest explanatory factors for FVC in the Tumen River cross-border basin (Figure 7). The explanatory power of both factors reaches 40%, indicating a significant impact and serving as the leading driving force. The next most significant driving factors are Ele (0.2446) and Slo (0.1184), which contribute to the spatial variation of FVC. Other factors such as Light, Tem, Lst, and Pre also have a certain influence on FVC, with Q Values exceeding 0.05, while Asp has the weakest explanatory power. Both

sides of the Tumen River basin have similar dominant driving factors for FVC changes, which are Pop and LULC. However, the explanatory power of LULC on the North Korean side is higher than that of Pop. Light was added as an important driving factor on the Chinese side, and the explanatory power of Ele, Light, and Slo increased compared to that of the whole basin. Among them, Ele has the highest Q Value of 0.3258, indicating a significant impact. The Q Values of Pre, Tem, and Lst on the Chinese side are 0.0877, 0.0855, and 0.0844, respectively. This indicates that the impact of meteorological factors on the change of FVC on the Chinese side is higher than that on the North Korean side.

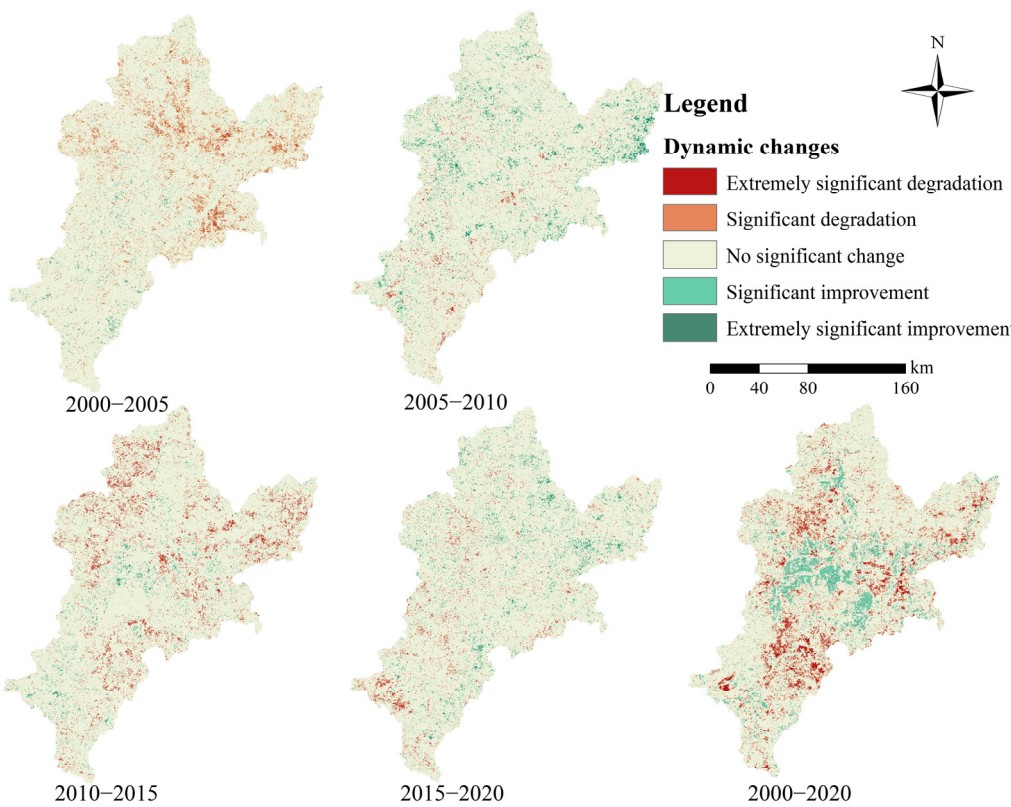

**Figure 6.** Dynamic changes of FVC in the Tumen River cross-border basin from 2000 to 2020.

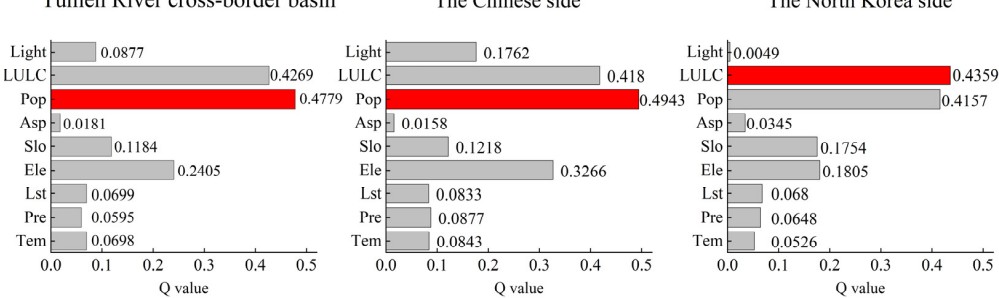

**Figure 7.** Factor detector results.

Various evaluation indicators have varying impacts on FVC, with differences observed in FVC between different ranges or types for a single indicator. By utilizing the risk detector, the suitable range for vegetation growth in the Tumen River cross-border basin was determined (Table 4). For climatic conditions, Tem ranged from 2.91–3.42 °C, Lst ranged from 7.437–7.443 °C, and Pre was between 661–675 mm, all of which resulting in high FVC. Topographical factors such as Ele ranging from 666 to 824 m, Slo between 12.4 to 17.5°, and Asp situated in the west have the largest average FVC. The vegetation conditions on both the Chinese and North Korean sides are also better with higher Ele and Slo values.

The three evaluation factors of LULC, Pop, and Light demonstrate the significant influence of human activities. Figure 3 shows that land cover type has a significant impact on vegetation, and woodland has the largest average FVC due to dense vegetation. Pop and Light values are low in areas with a low amount of human activity, resulting in higher FVC.

**Table 4.** Appropriate ranges or types for different factors.

| Evaluation Index | Vegetation Coverage Suitable Range | | | Mean Vegetation Coverage | | |
|---|---|---|---|---|---|---|
| | Tumen | China | North Korea | Tumen | China | North Korea |
| LULC | Forest | Forest | Forest | 0.710 | 0.732 | 0.654 |
| Tem | 2.91–3.42 °C | 2.98–3.42 °C | 4.66–5.2 °C | 0.713 | 0.754 | 0.612 |
| Pre | 661–675 mm | 670–679 mm | 624–636 mm | 0.693 | 0.736 | 0.653 |
| Lst | 7.437–7.443 °C | 7.44–7.45 °C | 7.61–7.63 °C | 0.760 | 0.755 | 0.662 |
| Ele | 666–824 m | 761–890 m | 1430–1640 m | 0.746 | 0.777 | 0.681 |
| Slo | 12.4–17.5° | 11.3–15.8° | 7.62–17.5° | 0.735 | 0.791 | 0.681 |
| Asp | West | West | Northwest | 0.676 | 0.716 | 0.616 |
| Pop | 0–0.0526 | 0–0.0526 | 0–0.789 | 0.773 | 0.774 | 0.736 |
| Light | 0–0.048 | 0–0.0482 | 0–0.0002 | 0.643 | 0.686 | 0.553 |

Note: The principle of maximizing the Q value guides the discretization of continuous variables in geographical detection. The appropriate range is determined by selecting the type with the largest average FVC after discretization.

The change in the FVC pattern is not the result of a single index factor but rather the product of the interaction of multiple factors. The results obtained from detecting the interaction of each evaluation factor (Figure 8) indicate that there are interactions between every factor in the Tumen River cross-border basin. The explanatory power of the interaction of different factors showed two-factor enhancement or non-linear enhancement, implying that the influence of the interaction factors was greater than that of a single factor. However, from the North Korean side, the combined effects of LULC and Light and Slo and Light weakened the explanatory power regarding FVC. In terms of the entire study area, the interaction influence of Pop and LULC ranked first, with a Q value of 0.5609, which was followed by the combination of Pop and Ele, with a Q value of 0.5468. The strongest influence on the Chinese side was the combination of Pop and Ele, with a Q value of 0.5622. Although the influence of the Lst single factor was weak, the combination of Pop and Lst ranked second with an influence of 0.5380, indicating that Lst is an indirect factor affecting the temporal and spatial distribution of FVC. The combination with a stronger interaction on the North Korean side is consistent with the characteristics of the entire basin, showing that the interaction between Pop, LULC, Ele, and other factors in the Tumen River cross-border basin still dominates FVC.

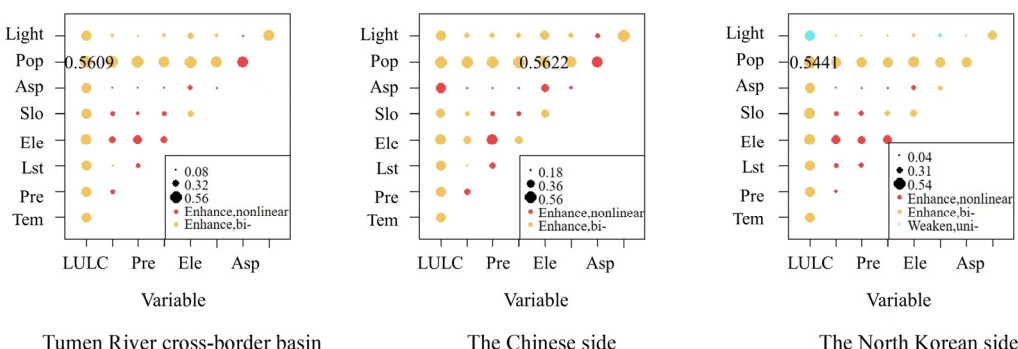

**Figure 8.** Driving factor interaction probe results.

*3.3. Habitat Quality Assessment Results*

In this study, the InVEST model was employed to conduct a preliminary assessment of HQ in the Tumen River cross-border basin from 2000 to 2020 (Figure 9). The results provide

a comprehensive insight into the impact of LULC type change on biological habitats. The HQ index of the basin is mainly composed of high-value areas ranging from 0.7 to 0.8, followed by low-value areas ranging from 0.2 to 0.3. The high-value areas are mainly distributed in the forested regions, whereas the low-value areas are concentrated in areas affected by high-intensity human activities such as cultivated land and artificial surfaces.

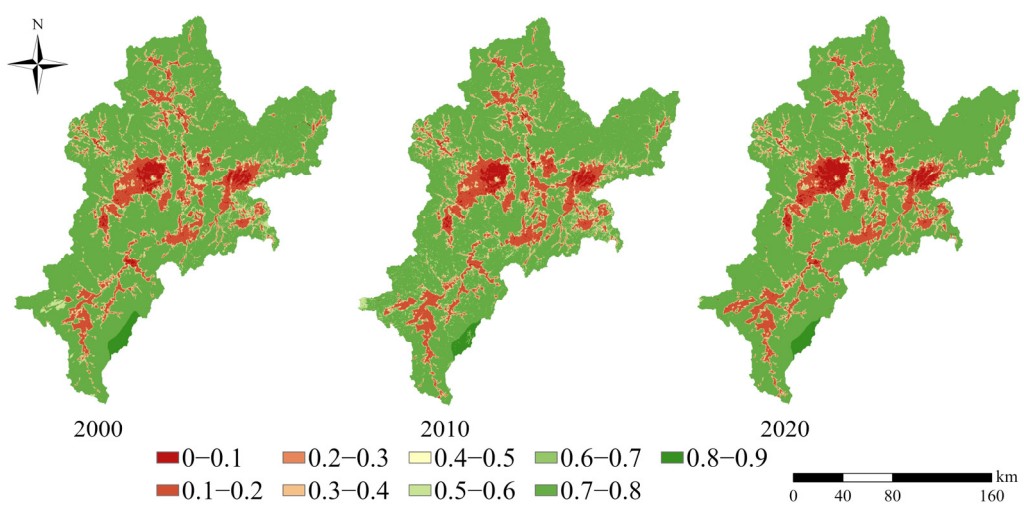

**Figure 9.** Spatial distribution of HQ Index based on LULC in the Tumen River cross-border basin from 2000 to 2020.

Due to the significant spatio-temporal variation in FVC in the Tumen River cross-border basin between 2000 and 2020, this study used the GWR tool to further explore the impact of FVC on HQ and its spatial differentiation characteristics (Figure 10). The results revealed clear spatial differences in the impact of FVC on HQ, with the regression coefficients showing a trend of high values in the surrounding areas and low values in the middle, which is a finding that is consistent with the spatial distribution of FVC in the basin. The regression coefficients ranged from 0.090 to 0.934, with an average of 0.606 and predominantly in high-value intervals of 0.690–0.768 followed by 0.605–0.690. These findings indicate that the degree of surface FVC has a significant positive impact on the HQ of the basin. Notably, areas with high FVC showed a high level of correlation between FVC and the HQ index, while low-value areas were mostly found in the middle of the study area, where FVC is lower. Therefore, this study concludes that the HQ index based solely on LULC assessment may not accurately reflect the HQ level of the entire basin.

The spatial distribution of the HQ index coupled with FVC in the Tumen River cross-border basin from 2000 to 2020 (Figure 11) reveals the strong explanatory power of FVC with respect to the pattern of the HQ index. The HQ index in the basin displayed a north–south gradient, sloping from northwest and northeast to southwest, with higher values in the north and lower in the south. The forest area exhibited a significant spatial difference in terms of the HQ index, which closely followed the spatial distribution pattern of FVC. Over the 20-year period, the HQ index and FVC of the basin displayed a relatively consistent change trend. During this period, the HQ index displayed a downward trend. The average value of the HQ index was 0.467 in 2000, which dropped to 0.447 in 2010. Compared to 2010, the HQ index in 2020 showed a slight decline, changing from 0.447 to 0.444. The change in the HQ index between 2010 and 2020 was minor compared to that in the preceding decade.

To describe the spatial pattern and aggregation characteristics of the HQ index in the Tumen River cross-border basin, this study employed Moran's I method to quantify the HQ index (Figure 12). The results show that the Global Moran's I value of the HQ index is positive and high, indicating a significant positive spatial correlation. From 2000 to 2010, the Global Moran's I value dropped from 0.648 to 0.628, indicating the weakened spatial agglomeration of the HQ index. The Global Moran's I value rose to 0.632 in 2020, showing

increased spatial autocorrelation. The scatter points of the HQ index mainly occupy the third quadrant, suggesting an obvious clustering of low values. The number of scatter points in the fourth quadrant increased, indicating an increase in areas surrounded by low-HQ-index areas, leading to a weakening of spatial agglomeration.

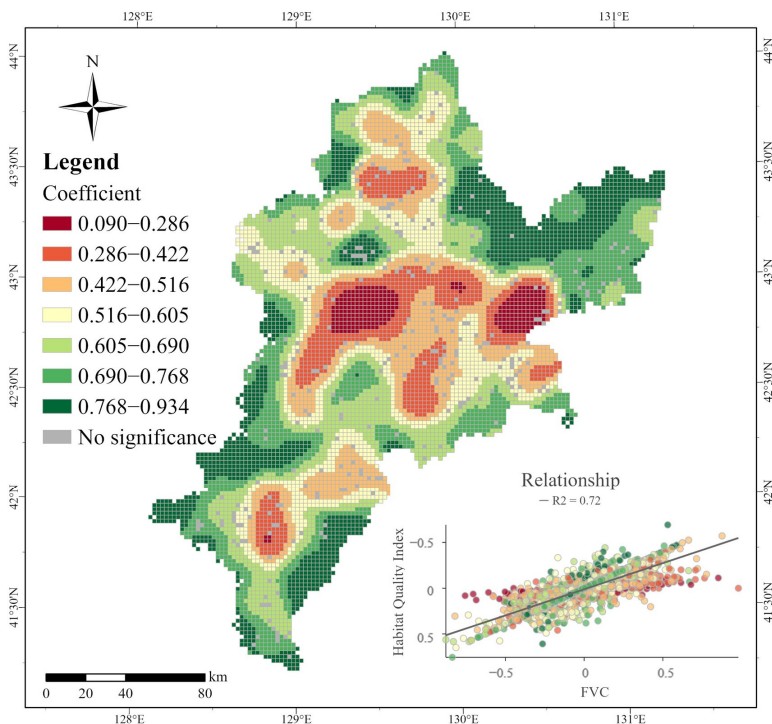

**Figure 10.** Spatial distribution of regression coefficients of HQ index and FVC in GWR model. The area where the residual value is greater than 2.5 and less than −2.5 was deemed insignificant.

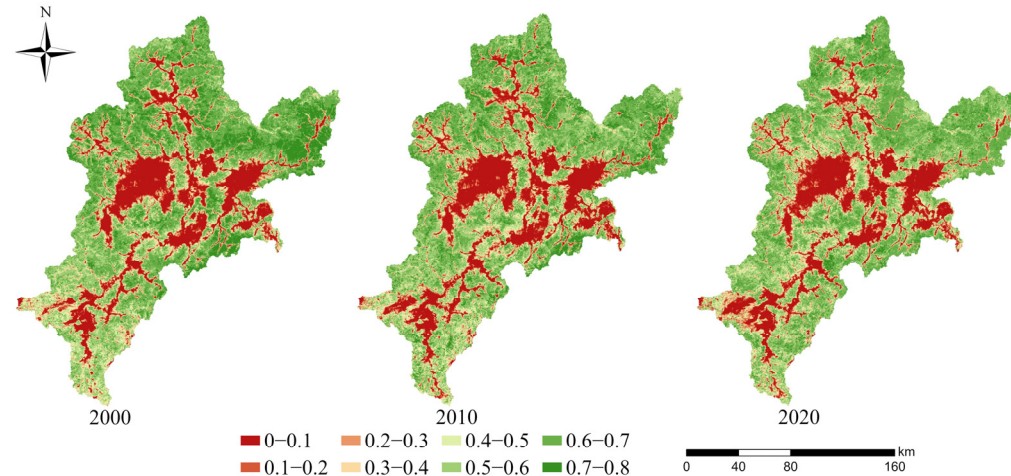

**Figure 11.** Spatial distribution of HQ index coupled with FVC in the Tumen River cross-border basin from 2000 to 2020.

The degree of spatial agglomeration of the HQ index in the Tumen River cross-border basin was assessed using the Global Moran's I method. To identify the specific locations of the agglomeration centers, Local Moran's I was further applied to analyze the HQ index (Figure 13). The analysis revealed that the basin primarily exhibits high–high and low–low spatial connections. The low–low clustering was mainly found in the central and southwestern parts of the basin, primarily in the areas of artificial surfaces and cultivated lands with high human activity intensity. On the other hand, the spatial distribution of

high–high agglomeration reflects the influence of FVC on HQ. The area of overlap between the distribution of high FVC and the high–high agglomeration is relatively significant and mainly located in the mountainous regions on the northern, eastern, and southwestern parts of the Chinese side. In general, the high–high agglomeration areas are mainly situated in regions with an exceptionally high HQ index, while the low–low agglomeration distribution is associated with areas with an exceedingly low HQ index.

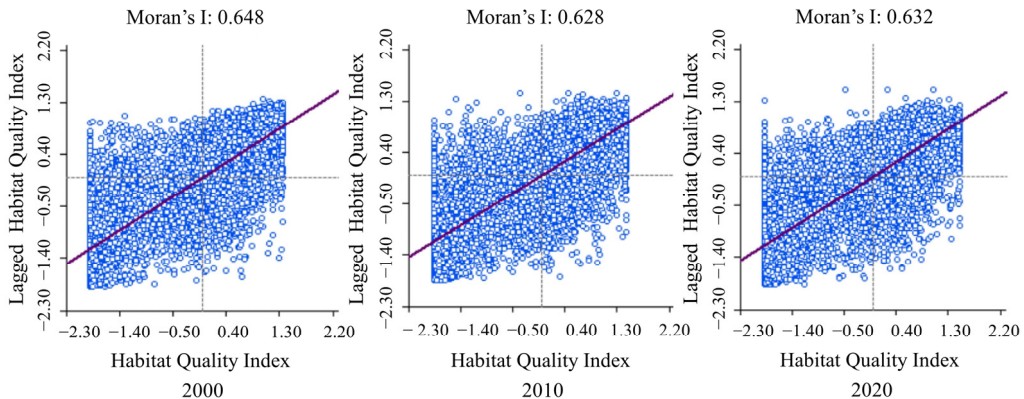

**Figure 12.** Global Moran's I scatter diagram of HQ index coupled with FVC in the Tumen River cross-border basin.

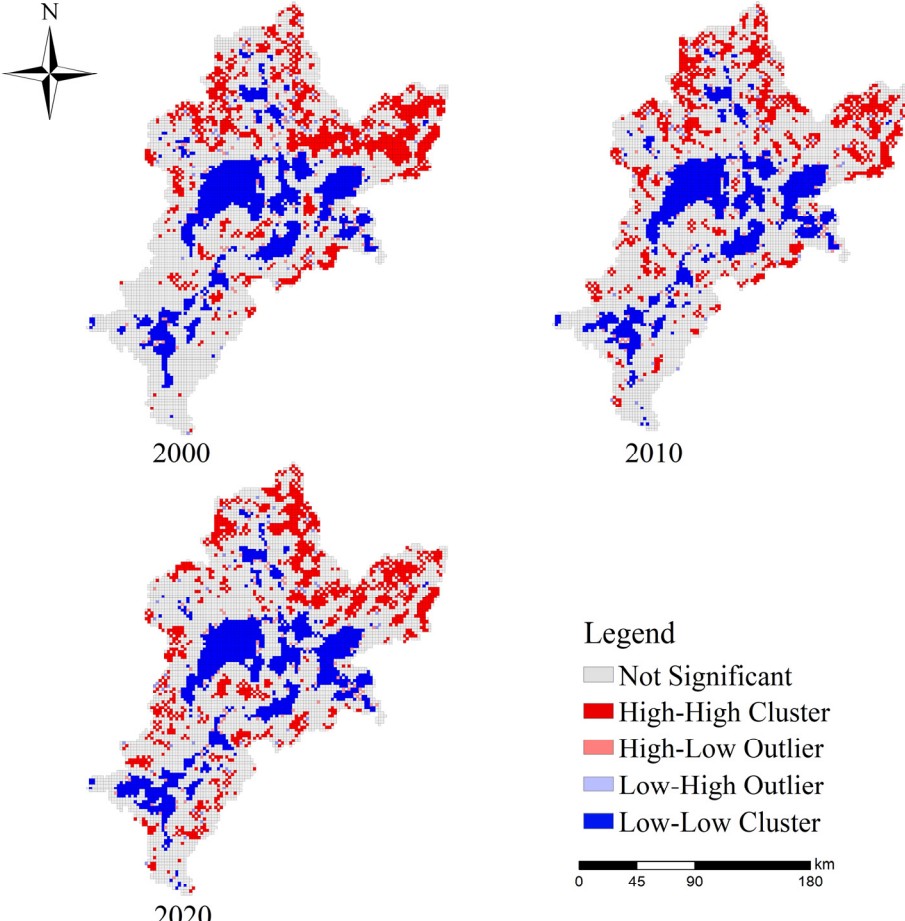

**Figure 13.** Cluster distribution of HQ index values coupled with FVC in the Tumen River cross-border basin from 2000 to 2020.

## 4. Discussion

### 4.1. Analysis of Spatio-Temporal Dynamic Variation of Vegetation Coverage

Based on long-term vegetation index data of the Tumen River Basin, this study analyzed the temporal and spatial dynamics of FVC using trend analysis and other methods. The results show that low values were recorded in 2005 and 2019, with the former being attributed to a large-scale summer drought and the latter being significantly impacted by forest fires on the Sino-Russian border. From a temporal perspective (Figure 2), the metabolic balance of vegetation was disrupted by the 2000–2005 summer drought, which seriously affected plant growth and survival, resulting in a significant decline in FVC. Between 2005 and 2010, relatively few extreme weather events occurred, leading to an increase in FVC. However, the 2010 major flood disaster disrupted vegetation restoration, and the FVC level of 2000 was not attained. From 2010 to 2020, the expansion of human activities and frequent extreme weather events resulted in relatively large interannual FVC variations. Overall, FVC exhibited a declining trend from 2000 to 2020, with approximately 11.94% of areas experiencing a decreasing trend.

In terms of spatial distribution and variation (Figures 3–6), the FVC on the Chinese side of the Tumen River cross-border basin is higher than that on the North Korean side. Low FVC values are concentrated mainly in the central and eastern parts of the basin, while high values are concentrated in the north and west. The project of "returning farmland to forests" has resulted in vegetation improvement areas being mainly distributed in the middle of the study area, mostly surrounding cultivated land. Initially, population density has been low in the northwest, southeast, and northeast mountainous areas of the study area, leading to good vegetation growth. However, with the expansion of human activities, a large amount of land has been destroyed, resulting in the degradation of FVC. To further validate the reliability of the vegetation coverage in the study area, this study conducted a comparative analysis between field measurement data and satellite observation data, revealing obvious similarities (Figure 14).

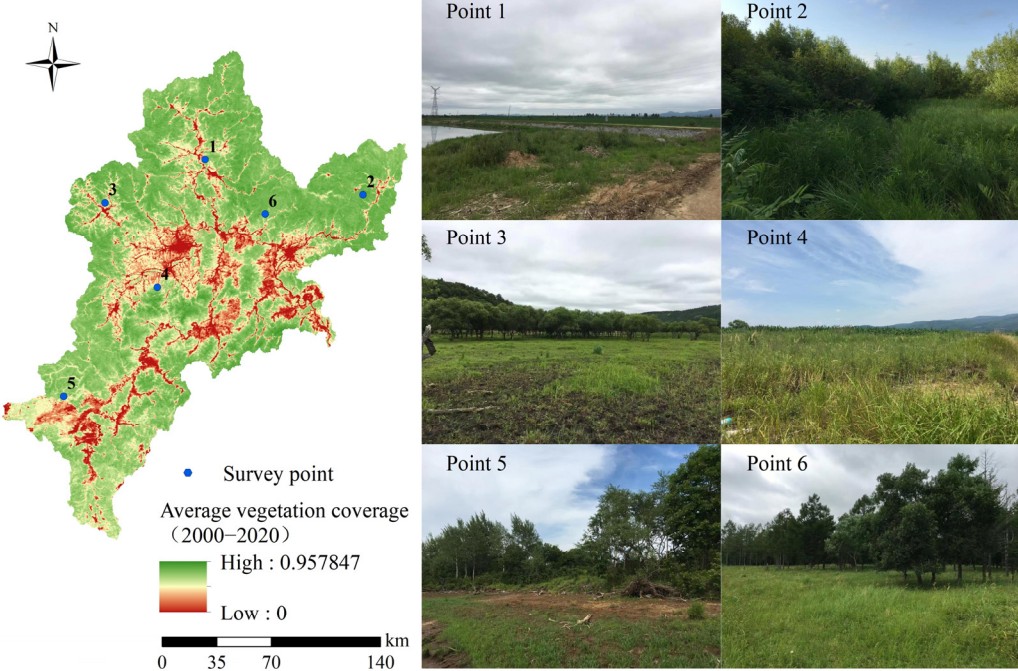

**Figure 14.** Field investigation of vegetation coverage in the Tumen River cross-border basin (accessed in July 2022).

### 4.2. Effects of Different Driving Factors on Fraction Vegetation Coverage

The interplay between human activities and natural factors has led to changes in the pattern and dynamics of land surface vegetation. The study area's driving force with respect to FVC (Figure 7) shows that human activities have a significant impact on FVC differentiation, which is consistent with previous research findings [67]. Changes in land use intensity resulting from alterations in land use types and population density are the primary ways in which human activities affect vegetation coverage. Among the various land use types, forest land has the highest FVC at 0.71, followed by grassland. Human beings use the artificial surface most intensively, resulting in the worst vegetation conditions for this land, with an FVC value of only 0.18, except for the low FVC caused by the water body's own characteristics. Additionally, there is a significant negative correlation between FVC and population density, fully reflecting the impact of land use intensity on FVC.

Changes in vegetation coverage patterns are also influenced by government policies. The Tumen River Basin, situated in the cross-border area between China and North Korea, is subject to policy differences between the two countries, which play a significant role in the alteration of FVC. The impact of policies on FVC is reflected in many aspects, including changes in agricultural production structures due to policy intervention and the multi-directional expansion of rail transit under state investment and construction.

The economic development of the Chinese side of the study area has been greatly facilitated by national policies, which has led to accelerated regional development. As a result, large amounts of forest and grassland in the region have been developed into cultivated land, and the forest itself has also been destroyed, which has led to a decline in FVC. According to the China Forestry Statistical Yearbook (Figure 15), China's forest development has continued to expand from 2000 to 2020, with the total value of the forestry industry increasing by CNY 7816.37 billion. This trend is also evident on the Chinese side of the Tumen River cross-border basin, where the FVC of forested area has dropped significantly from 0.757 to 0.725. In addition, China has introduced sustainable development policies such as the "returning farmland to forest" and "returning farmland to grassland" pledges. As a result, the vegetation restoration area on the Chinese side of the study area from 2000 to 2020 was mostly located around cultivated land. On the other hand, the North Korean side, guided by the national principle of "self-reliance", has engaged in the large-scale deforestation and reclamation of hillside terraces to ensure food security, leading to significant damage to surface vegetation. As a result, the FVC on the North Korean side of the study area is significantly lower than that on the Chinese side.

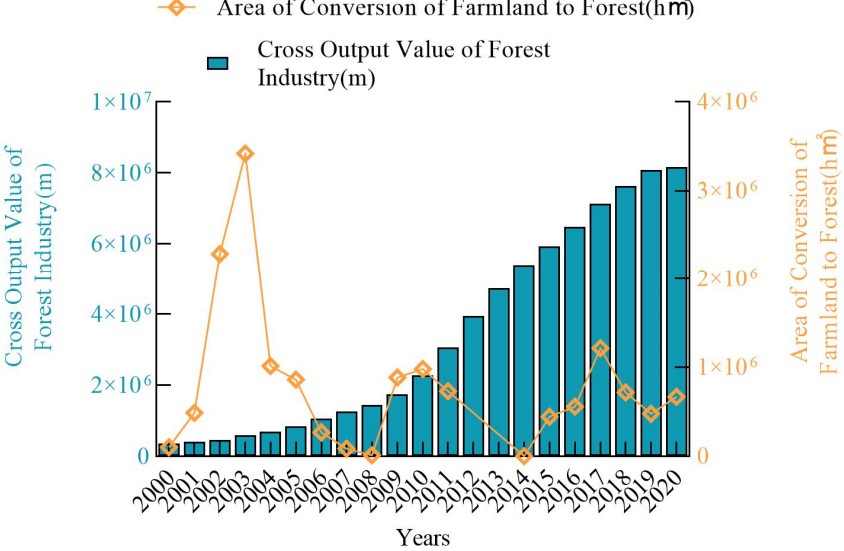

**Figure 15.** The area of farmland converted to forests and the cross-output value of the forest industry in China from 2000 to 2020.

Topographic factors play a crucial role in redistributing water, heat, and nutrients between regions via different processes, thereby affecting regional vegetation patterns [68]. In the findings of this study, the areas with the highest FVC are situated at elevations between 666 and 824 m (Table 4). Moreover, the FVC shows an increasing trend with an increase in slope. This trend could be attributed to the fact that frequent human activities on the horizontal terrain have inhibited vegetation growth. With an increase in elevation and slope, the influence of human activities weakens, and the vertical differentiation of the climate becomes more pronounced, creating an environment that favors vegetation growth and development. However, with a further increase in elevation, the FVC shows a declining trend due to factors such as a gradual decrease in temperature, a lack of minerals in the soil, and excessive sunlight intensity.

The Tumen River cross-border basin is situated in the mid-to-high latitude region, with the latitude on the Chinese side being relatively high, rendering vegetation more susceptible to climate factors. In recent years, the concentration of greenhouse gases such as $CO_2$ and $CH_4$ in the atmosphere has continued to increase, leading to the acceleration of the rate of climate warming and the exacerbation of the unpredictable nature of climate change [69]. This trend has resulted in an increase in the inconsistency of the combination of water and heat conditions in the study area, which could have an impact on the growth and distribution of plants and, subsequently, affect FVC.

*4.3. Effect of Human Activities and Fraction Vegetation Coverage on Habitat Quality*

HQ represents a vital component of natural ecosystems that remains highly vulnerable to external disturbances, particularly those resulting from human activities [21,70]. In 1992, a collaborative effort between China, Russia, North Korea, South Korea, and Mongolia led to the initiation of the Tumen River Regional Cooperation Development Project. In recent years, international cooperative efforts in the Tumen River region have deepened, fostering rapid economic growth, urban expansion, and changes in agricultural practices. These developments have exerted escalating pressures on indigenous species and their habitats. Consequently, in future investigations focusing on HQ assessment, it has become imperative to comprehensively comprehend the impact of human activities, particularly national policies, on biodiversity and habitat preservation.

The impact of vegetation on HQ is primarily reflected in the influence of vegetation type, density, and structure on species habitat selection. Additionally, vegetation is a vital component of habitats, and the quality of vegetation reflects the quality of a regional habitat to a certain extent. Vegetation coverage data can be used to accurately monitor the dynamic changes of vegetation in time and space, enabling the quantification of the suitability level of the current habitat. This paper employed the GWR model, with the change in HQ assessed via the InVEST model acting as the dependent variable and the change in FVC as the independent variable, to quantify the degree of influence of FVC on HQ. This research demonstrates (Figure 10) that all regression coefficients are positive and predominantly concentrated in the high-value range of 0.690–0.768, indicating that the degree of FVC significantly promotes the HQ of the Tumen River cross-border basin. The spatial distribution of the correlation between FVC and the HQ index is generally consistent with the distribution pattern of FVC, indicating that the positive impact on the HQ of the basin deepens with the increase in surface vegetation coverage. It is evident that relying solely on land use classification data will not result in an accurate representation of the HQ of the basin.

*4.4. Habitat Quality Assessment Coupled with Fraction Vegetation Coverage*

This paper comprehensively and scientifically evaluated the HQ of the Tumen River cross-border basin using FVC as a metric. Changes in FVC affect species–habitat relationships and thus indicate the state of vegetation growth and the food abundance for species. As a result, variations in FVC have a significant impact on the HQ of a region. In contrast, alterations in land use types mainly reflect changes in the intensity of human activities,

which are crucial factors affecting species habitats. The intensity of human activities in residential areas, cultivated land, and road areas is potent, thereby posing a significant threat to organisms. Consequently, changes in this area are a vital contributor to the changes in the HQ index [71].

The FVC in the northern and surrounding areas of the Tumen River cross-border basin is significantly higher than that in the southern and central areas (Figure 3). Furthermore, according to the spatial distribution data of land use types, it can be inferred that the intensity of human activities in the central area of the basin is greater than that in the surrounding areas. The HQ index of the basin has a spatial pattern of high values in the north and low values in the south and central areas along with the center of the surrounding areas (Figure 11). Consequently, the land use types in low-value HQ index cluster areas are predominantly man-made surfaces and cultivated land, and the FVC in these areas is low. The spatial distribution of high HQ index values is related to areas with low human activity intensity or high FVC. If there is an overlapping area between the two to a certain extent, then high-HQ-index clusters will form in this area (Figure 13).

In chronological terms, the HQ index decreased in 2010 compared to 2000, and the average values of the HQ index in 2010 and 2020 were similar. Similarly, vegetation coverage in the study area decreased from 0.6438 to 0.6266 from 2000 to 2010, while the FVC in 2020 was 0.6234, which was similar to that in 2010. These results fully demonstrate the impact of FVC on HQ. A comparison of the evaluation results regarding HQ in this study with the results of the cross-border areas of China, North Korea, and Russia calculated using Gan.X based on the InVEST model [60] reveals a relatively consistent spatial distribution pattern and temporal variation characteristics. Zhang.Y et al. also evaluated the HQ on the Chinese side of the Tumen River cross-border basin and found that it improved slightly [72]. Although their results differ from this study due to the different time scales, the spatial distribution patterns of the HQ index are similar. Moreover, because FVC was used to correct the HQ, the overall HQ index was lower than the two results.

Given that the capacity of a given habitat to provide ecosystem services for species in various regions varies, FVC, as a manifestation of habitat quality, can be used to optimize HQ assessment. However, this approach simplifies the actual process, disregarding numerous interconnected factors and unknown mechanisms. Although it may not adequately represent local biodiversity levels, comprehensive HQ assessments provide insights into the capacity for regional biodiversity maintenance and the extent of disturbance. In future investigations, we will explore the integration of diverse vegetation indicators and other factors along with the incorporation of satellite observations [73], aiming to achieve a more precise evaluation of HQ.

## 5. Conclusions

This study analyzed the spatiotemporal variation pattern, driving factors, and impact on habitat quality of the long-term FVC extracted from the Tumen River cross-border basin. Furthermore, the study incorporated the FVC to improve the HQ assessment model. After analyzing and discussing the results, the following conclusions were drawn.

The Tumen River cross-border basin has a relatively high overall FVC level, which showed a downward trend from 2000 to 2020. The Chinese side of the basin has a higher overall FVC than the North Korean side. Low FVC values are mainly concentrated in the middle and eastern parts of the basin, while high FVC values are mainly concentrated in the north and west. FVC is significantly affected by human activities and topographic factors. Land use, population density, and elevation factors are the main driving factors of temporal and spatial changes in FVC, while climate factors have little influence. FVC has a significant impact on habitat quality patterns, especially in areas with stable land use. From 2000 to 2020, the HQ index of the coupled FVC in the basin showed a downward trend with significant spatial autocorrelation. High–high clusters mostly overlap with forest areas with extremely high FVC, while low–low clusters overlap with areas with extremely low FVC and high human activity intensity.

Based on the research results presented above, the following recommendations are proposed for the ecological management planning of the Tumen River cross-border basin: First, monitor and track changes in vegetation cover in the basin and strengthen its protection and management. Second, employ targeted measures to control the areas with low FVC and severe degradation while safeguarding the areas with good vegetation conditions. Third, consider ecological security when formulating and implementing policies, and mitigate the threat posed by human activities to natural habitats.

**Author Contributions:** Conceptualization, R.J. and Y.W.; methodology, D.Q. and Y.W.; software, Y.W.; validation, D.Q. and Y.W.; formal analysis, Y.W.; investigation, Y.W.; resources, R.J.; data curation, Y.W.; writing—original draft preparation, Y.W.; writing—review and editing, R.J., D.Q., W.Z. and Z.L.; visualization, Y.W.; supervision, R.J. and D.Q.; project administration, R.J.; funding acquisition, R.J. All authors have read and agreed to the published version of the manuscript.

**Funding:** This research was funded by the National Natural Science Foundation of China (41830643, 41807508), the National Science and Technology Fundamental Resources Investigation Project (2019FY101703), the Jilin Provincial Science and Technology Department Project (20210101106JC), and the Natural Science Foundation of Jilin Province of China (20200201044JC).

**Institutional Review Board Statement:** Not applicable.

**Informed Consent Statement:** Not applicable.

**Data Availability Statement:** No new data were created or analyzed in this study. Data sharing is not applicable to this article.

**Conflicts of Interest:** The authors declare no conflict of interest.

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
