# Peer review of "Habitat Quality Assessment under the Change of Vegetation Coverage in the Tumen River Cross-Border Basin"

_sustainability, doi:10.3390/su15129269_

Round 1

Reviewer 1 Report

This study used FVC derived from MOD13Q1 NDVI data, together with climatic and topographic data from 2000 to 2020 to analyze the impact of FVC on HQ. It’s quite valuable for related studies and practices. The manuscript is well structured, with sound methodologies and results.I suggest enhance the analyze in results and discussion Sections the impact of Human activities on HQ,not only the FVC in the current version.

Reviewer 2 Report

Comments to the manuscript by Wang et al.

This manuscript explored the habitat quality assessment under vegetation coverage change in the Tumen River cross-border basin.

The manuscript was written according to the results of hard work in remote sensing. It has a good idea as it aims to provide a scientific theoretical basis and data support for the ecological environment planning of the Tumen River Basin and the coordination of nature, economy, and society toward green and sustainable development.

It has a good introduction, methodology, results, and discussion structure. However, the study has a few references, especially in the discussion, methodology, and methods; the authors can add more references. Also, the authors can briefly describe the vegetation cover in the study area and give examples of dominant plant species.

I have one question: Does this type of study need ground truthing (collecting data from the field) besides remote sensing to confirm the results?

 Minor editing of the English language required, please

Reviewer 3 Report

Dear Authors,

I think that your manuscript is a rather careful and conscientious, also refined from the editorial point of view.

However, I have 2 main objections: 

1    Linking habitat quality with assessment under the Change of Vegetation Coverage, is undoubtedly not a bad idea, is not sufficient. FVC is only one of the possible predictors, so can not explain the entire relationship between the above-mentioned variables. In the introduction, it should be written more clearly about it, and in the conclusions, it should be emphasized as the weaker side of the work, possibly with plans for the future to supplement the results. As compensation, works related to the integration of satellite observations in environmental research should be cited, e.g. Pasetto D, Arenas-Castro S, Bustamante J, et al. Integration of satellite remote sensing data in ecosystem modeling at local scales: Practices and trends. Methods Ecol Evol. 2018;9:1810–1821.

2. Another important issue that needs to be clarified is that the authors use either "pixel-by-pixel" analyses or very traditional and limited spatial statistics. There are no geostatistical analyses, at all, which is also a weak side of the work. At this stage, it is difficult to expect and require that the authors include new analyses, but at least they should mention in the introduction, the need and possibility of such analyses, supplementing the literature with e.g. the following review papers.

Fatemeh Zakeri, Gregoire Mariethoz, A review of geostatistical simulation models applied to satellite remote sensing: Methods and applications, Remote Sensing of Environment,

259, 2021, 112381.

J. Zawadzki, C.J. Cieszewski , M. Zasada and R.C. Lowe, Applying geostatistics for investigations of forest ecosystems using remote sensing imagery. Silva Fenica. 2005. 39(4):599-617.

Su, H., Shen, W., Wang, J. et al. Machine learning and geostatistical approaches for estimating aboveground biomass in Chinese subtropical forests. for. Ecosystem 7, 64 (2020). https://doi.org/10.1186/s40663-020-00276-7.

Otherwise, the reader may get the impression that the analyzes are too narrow and oversimplified as well as that the authors do not know enough literature on the subject, even the review papers.

In summary, the above issues should be mentioned in the introduction, and in particular the work should be supplemented with a more balanced discussion about necessity of using integration (using also geostatistical methods)of different vegetation indices, and bibliography. The weak side of the work should be clearly underlined and discussed. 

                                           Sincerely yours,

                                                               Reviewer
